# Experimental and Finite Element Study on Bending Performance of Glulam-Concrete Composite Beam Reinforced with Timber Board

**DOI:** 10.3390/ma15227998

**Published:** 2022-11-12

**Authors:** Hao Du, Shengnan Yuan, Peiyang Liu, Xiamin Hu, Guohui Han

**Affiliations:** 1College of Civil Engineering, Nanjing Forestry University, Nanjing 210037, China; 2College of Civil Engineering, Nanjing Tech University, Nanjing 211816, China

**Keywords:** glulam-concrete composite beam, timber board interlayer, bending performance, finite element model, parametric study

## Abstract

In this research, experimental research and finite element modelling of glulam-concrete composite (GCC) beams were undertaken to study the flexural properties of composite beams containing timber board interlayers. The experimental results demonstrated that the failure mechanism of the GCC beam was the combination of bend and tensile failure of the glulam beam. The three-dimensional non linear finite element model was confirmed by comparing the load-deflection curve and load-interface slip curve with the experimental results. Parametric analyses were completed to explore the impacts of the glulam beam height, shear connector spacing, timber board interlayer thickness and concrete slab thickness on the flexural properties of composite beams. The numerical outcomes revealed that with an increase of glulam beam height, the bending bearing capacity and flexural stiffness of the composite beams were significantly improved. The timber boards were placed on top of the glulam members and used as the formwork for concrete slab casting. In addition, the flexural properties of composite beams were improved with the increase of the timber board thickness. With the elevation of the shear connector spacing, the ultimate bearing capacity and bending stiffness of composite beams were decreased. The bending bearing capacity and flexural rigidity of the GCC beams were ameliorated with the increase of concrete slab thickness.

## 1. Introduction

The glulam-concrete composite (GCC) beam is a novel structural element in which the glulam member and the concrete slab fit together through shear connections. Due to the sufficient composite effect provided by shear connections, the material properties of the glulam and concrete are better utilized, as tensile forces from bending are primarily resisted by the glulam and compressive forces are resisted by the concrete [1,2]. The GCC beam exhibits better bending, bearing capacity and flexural rigidity than the pure glulam beam, and displays greater acoustic insulation, better vibration comfort, and superior fire protection ability [3,4,5]. Over the past ten years, the GCC beam has been extensively utilized to reinforce and strengthen timber slabs in existent and newly-built multistory buildings. In different practical uses of GCC beam, especially in the restoration of timber architectures, the timber boards are placed on top of glulam members and can be used as the formwork for concrete slab casting [6].

Up to now, substantial experimental studies on the structural behaviors of GCC beams have been published [7,8,9,10]. Clouston et al. [11] performed bend tests on GCC beams with steel toothed plate connectors. It was verified that composite beams exhibited excellent mechanical properties. Persaud et al. [12] conducted experimental studies on the flexural properties of GCC beams with screw connectors. The outcomes revealed that the flexural bearing capacity and rigidity of GCC beams were significantly higher than those of pure glulam beams. Khorsandnia et al. [13] completed experimental research on the short-term behavior of GCC beams with different forms of shear connectors (lag screws, SFS and triangular notched connectors). Through nonlinear regression, analytical models were established for investigating the load-slip behavior of the three types of connections. Schanack et al. [14] found that during the stress process of GCC beams with screw connectors, the cracking of concrete slabs reduced the stiffness of concrete, which resulted in the increase of the deflection deformation and section stress of composite beams. Hong et al. [15] conducted bending tests to investigate the effects of shear connection ratio and concrete type on the bending performance of GCC beam. The outcomes revealed that the failure modes of the GCC beam were tensile brittle failure and finger joint failure of the glulam beams. Giv et al. [16] used adhesive to connect the concrete slab and glulam beam, and carried out experimental research and theoretical analysis pertaining to the shear bonding performance of the adhesive and the flexural performance of the composite beams. Du et al. [17] conducted experimental research on the fire behavior of GCC beams with timber board to study the effects of the timber board thickness and load ratio, and established the mechanical model for analyzing the flexural bearing capacity and rigidity of GCC beams in the presence of fire.

There are multiple numerical research and theoretical analyses regarding the structural behavior of GCC beams. The existent finite element (FE) models for GCC beams can be separated into 1D, 2D, and 3D. Lukaszewska et al. [8] established one-dimensional FE models of timber-GCC floors. The glulam beam and concrete slab were emulated as two parallel beams, and the shear connectors were simulated by discrete nonlinear spring elements. The shear lag effects on the concrete slab were ignored, which might induce computational outcome overestimation when the broad slab was utilized. Crocetti et al. [18] developed two-dimensional FE models for timber-concrete composite beams produced via simulating the concrete slab with multi-layer shell elements and emulating the timber beam by beam elements. The discrete steel reinforcement bars were assumed as the steel layers with the identical section area, and the interface connections were defined by the non-linear springs. Thereby, the shear lag effects on concrete slabs and the impairment plasticity behaviors of concrete were covered by the model. The research conducted by Monteiro et al. [19] proved that two-dimensional FE models exhibited remarkable accuracy based on the test outcomes, particularly for composite beams with comparatively larger slab widths. Nevertheless, given that timber was a kind of orthotropic material, neglecting the effect from weak orientations could cause the slight overestimation of the bending performances of the GCC beam. To thoroughly reveal the influence of material performance and the interplay between the members, solid elements with the specific definitions of the stress-strain and contact associations were used in three-dimensional FE models. The studies validated that three-dimensional FE models could precisely describe the load-deflection, stress distributional status, and slip behavior of timber-concrete composite beams [20,21,22].

Up to now, few researches on the structural performance of GCC beams containing timber board interlayers have been completed. Here, the experimental research on glulam-concrete composite beams was undertaken to study the flexural properties of composite beams containing timber board interlayer. Then, the three-dimensional non-linear finite element model was developed and confirmed by comparing the load-deflection curve and load-interface slip curve with the experimental results. Based on the finite element studies, the influences of the glulam beam height, shear connector spacing, timber board interlayer thickness and concrete slab thickness on the flexural properties of composite beams were investigated.

## 2. Bending Tests

### 2.1. Composite Beam Sample

One full-scale GCC beam with inclined fasteners was constructed and loaded by the four-point bending approach. In the composite beam sample, the glulam beam was connected to the concrete slab using shear connectors (Figure 1). The glulam beam was 150 mm wide, 300 mm high and 3900 mm long. The concrete slab was 80 mm thick and 800 mm wide. The concrete slab was enhanced with steel meshes utilizing 8 mm reinforcement bars with a spacing of 150 mm in two orientations. A timber board was placed on the top of the glulam beam and utilized as formwork for concrete slab casting. The inclined screw exhibited higher shear bearing capacity and slip modulus than that of screws inserted vertically. Thus, an overall 19 horizontal cross-wise pairs of inclined screw fasteners were used for connecting. The embedment lengths into concrete and timber of the screw connectors were 70 and 100 mm, respectively. The glulam beam comprised evenly glued laminated larch timber. The mean density of the timber was 0.526 g/cm^3^, and the mean water content was 12.8%. The compressive elasticity modulus and compressive strength of the lumber were obtained through material tests according to ISO 3787 [23]. The average compressive elastic modulus was 11,580 N/mm^2^, the compressive strength parallel to the timber grain was 39.6 N/mm^2^, and the tensile strength parallel to the timber grain was 84.89 N/mm^2^. The mean compressive strength of the concrete was 35.8 N/mm^2^.

The four-point bending loads were applied to the beam specimen using a 300 kN hydraulic jack. Firstly, the applied loads were applied at a rate of 0.2 kN/s to 10% of the anticipated ultimate load, and then the loads were released to verify the efficiency of the bending test setup and measurement equipment. After that, the load increased at a rate of 0.2 kN/s until it reached the ultimate load. For obtaining the relative slips between the concrete slab and the glulam beam, two displacement meters were mounted at the end of the GCC specimen.

### 2.2. Test Results

At the beginning of loading period, there was no observable relative slip because of the strong bonding effect at the interfacial region. With an increased level of the applied load, a small relative slip was identified. For the composite beam sample, when the applied load was elevated to approximately 85% of the ultimate load, cone expulsion failure happened on the concrete layer around inclined screw fasteners (Figure 2a). The total collapse happened on the testing sample because of the combination of bend and tensile failure in the glulam beam (Figure 2b). Figure 3a displays the association between the loads and mid-span deflections acquired from the bend test. The design of the glulam-concrete beam is usually controlled by deflection. The corresponding mid-span deflection under serviceability limit state Δ_SLS_ was based on GB 50,005 [24], Δ_SLS_ = *l*/250. At the beginning of loading, the GCC beam displayed linear elasticity behavior. When the load was elevated to about 60% of the maximum load, the applied loads presented an analogous parabolic increase with the elevation of the mid-span deflections. The flexural stiffness of GCC beam was lower, and diminished continuously. When it attained the maximal load, with the elevation of deflections, the loads decreased remarkably because of the brittle failure of the glulam beam. Figure 3b presents the association between the loads and interface slips at the end of the GCC beam. With the bending loading elevated, the relative slips at the at the interfacial region increased because of the bending deformation of screw fasteners under longitudinal shearing force.

## 3. Finite Element Model

### 3.1. Constitutive Model of Materials

To sufficiently consider the effect of the failure mechanisms on the flexural behavior of GCC beams containing interlayers, three-dimensional (3D) FE models were established by the commercially available FE software ABAQUS. In the FE models, the damage plasticity model was utilized to realize the modeling of the physical performances of concrete. The concrete damage elicited by cracking or crushing causes a decrease in the elasticity modulus of the concrete. The tensile behavior of concrete was hypothesized to be linear up to the uniaxial tensile strength equal to *f*_t_ = *f^’^*_c_/10. According to Eurocode 2 [25], the compression uniaxial stress-strain association for concrete can be identified by the formula below:(1)$$\frac{\sigma_{c}}{f_{\mathrm{cm}}}=\frac{k\eta -\eta^{2}}{1+(k-2)\eta }$$
where: *η* = *ε*_c_/*ε*_c1_; *ε*_c1_ = the strain at peak stress; $k=1.05E_{cm}\times \left|\varepsilon_{c1}\right|/f_{\mathrm{cm}}$; *E*_cm_ = elasticity modulus of concrete; *f*_cm_ = cylinder compression strength of concrete.

The timber was simulated as an orthotropic material. It was hypothesized that both rigidity and intensity in the radial or tangential orientations were identical. The association between elasticity modulus in the grain and cross grain orientation as well as the association between elastic modulus and shearing modulus are specified in standard EN 338 [26]. The orthotropic yield standard put forward by Hill [27] was utilized to realize the definition of glulam, which has been evidenced to have precise delineation of timber performances [28].

The steel material properties of screw fasteners were defined as isotropic elastic-plastic constitutive. The yield strength and ultimate tensile strength of screw fasteners were 375 MPa and 462 MPa, which were measured from material tests.

### 3.2. Element Type and Interface Simulation

The composite beam specimen was mainly composed of the concrete slab, glulam beam, screw fasteners and steel bars. The screw section cutting of the threaded section was considered in the screw modeling. The net section diameter was used for the screw modeling. The 8-node solid linear elements with reduced integration C3D8R were utilized to model the concrete slab, glulam beam and screw fasteners. The T3D2 truss element was utilized to simulate the steel reinforcement in the concrete slab. The global size of the element meshed in the connection model was 10 mm, and for the beam model, the relevant size was 100 mm. The interplay between the steel bars and concrete were analyzed through the ‘‘Embedded region” option. The longitudinal spacing and transverse spacing of screw fasteners were 200 mm and 50 mm, respectively. Both tangential and normal contact behaviours were identified in the FE model to simulate interaction among screw fasteners, concrete slab and glulam beam. The option ‘‘Hard Contact” was selected to identify the normal contact behaviours, and ‘‘Penalty” was chosen for identifying tangential behaviourd through postulating the timber-concrete frictional coefficient of 0.7. The model diagram of the GCC beam after defining boundary and load conditions is shown in Figure 4.

### 3.3. Validation and Discussion

Figure 4 displays the stress distributional status of the GCC beam at the maximal load. When the composite beam model reached the limit state, the midspan bottom of the glulam beam achieved the ultimate tensile stress (Figure 5a). In addition, the maximum principal stress distribution of the concrete slab indicated that the cone expulsion failure occurred at the inclined screw fasteners (Figure 5b). The outcomes revealed that the numerical simulation could precisely predict the failure mode and the change characteristics of the composite beam in the loading process. Figure 6 presents the numerical simulation results of the load-deflection relationship and load-interface slip curve for the GCC beam. As shown in Figure 5, the developed finite element model adequately predicted the load-deflection relationship and load-interface slip curve, particularly in the elasticity range. With the elevation of the applied load, the cracks appeared on the weak surface of the glulam member under the combination of bending and shear effects, which resulted in reduction of the stiffness and load carrying capacity. The maximal bearing capacity of the composite beam forecasted by the finite element modelling was a little lower than the testing outcomes. The main reason was the simplified constitutive law of timber utilized in the finite element modelling. At the beginning of loading, the slip displacement was small because of the strong bonding effect at the concrete-timber interfacial region. When the loading was elevated to approximately 28% of the maximum load, with the applied loads increased, the attenuated composite actions at the interfacial region led to a rapid increase of the interface slips.

## 4. Parametric Study

### 4.1. Section Height of Glulam Beam

To reveal the impact of the glulam beam section size on the flexural performance of composite beams, different sizes of the glulam beams (150 mm × 300 mm, 150 mm × 200 mm, 150 mm × 100 mm and 150 mm × 400 mm) were simulated. The numerical simulation results of the composite beam with different glulam beam section size are presented in Figure 7. When the section height of the glulam beam increased from 300 mm to 400 mm, the elastic bending capacity of the GCC beams increased by 51.4% and the ultimate bearing capacity increased by 59.4%. When the section height decreased from 300 mm to 200 mm, the elastic bearing capacity was reduced by 42.9% and the maximal load carrying capacity was reduced by 38.2%. The comparison results showed that the section height of the glulam beam could significantly affect the flexural bearing capacity. Through the analysis of the load-deflection curves, it was found that the flexural rigidity of the composite beam was improved with the increase of the glulam beam section height. In addition, the increasing section height of the glulam beam could significantly ameliorate the deformation-resistance capability of GCC beam.

### 4.2. Screw Connector Spacing

The shear connections play an important role in GCC beams through transferring shear forces among the glulam and concrete members and preventing detachment mechanisms. Four finite element models of the GCC beams with 100, 200, 300 and 400 mm spacing of inclined screw fasteners were simulated and analyzed to research the influence of shear connector ratio on the flexural properties. The load-deflection curves of GCC beams with different shear connector spacings are displayed by Figure 8. The numerical results revealed that the ultimate bending bearing capacity of the GCC beams with 100, 200 and 300 mm spacing of screw connectors increased by 13.5%, 7.6% and 4.0%, respectively, compared to that of the composite beam with 400 mm spacing of screw connectors. In addition, the flexural stiffness of the GCC beams was improved by 6.3%, 20.0% and 33.6%, respectively, compared to that of the composite beam with 400 mm spacing of screw connectors. Thus, the bending bearing capacity and flexural stiffness of the composite beams decreased with the increase of the screw fasteners spacing, which was mainly due to the decrease of the composite action between the glulam beam and the concrete slab.

### 4.3. Thickness of Timber Board

Timber boards may be utilized as the permanent formwork for concrete casting. This can not only create convenience for the construction, but also keep the integral beauty of the timber structures. Five finite element models of GCC beams with various timber board thicknesses were established to study the impacts of timber board thickness on the structural properties. The load-deflection curves of the GCC beams are presented in Figure 9. The numerical simulation outcomes revealed that when the timber board thickness increased from 0 mm to 10 mm, the flexural stiffness and ultimate bending bearing capacity of the composite beams displayed no obvious difference, which indicated that the presence of the timber board had no significant effect on the mechanical performance of GCC beam. The maximal bending bearing capacity of the GCC beams with 15, 20 and 30 mm timber board thickness was improved by 3.2%, 7.5% and 10.8%, respectively, compared to that of the composite beam with 10 mm timber board thickness. It was discovered that the bending bearing capacity and flexural stiffness of the composite beams was enhanced with the elevation of the timber board thickness.

### 4.4. Thickness of Concrete Slab

Five finite element models of GCC beams with various concrete slab thicknesses were established to study the impact of concrete slab thickness on the mechanical properties. The load-deflection curves of the GCC beams with 60, 70, 80, 90 and 100 mm concrete slab thickness are displayed in Figure 10. The numerical simulation outcomes revealed that the ultimate bending bearing capacity of the GCC beams with 70, 80, 90 and 100 mm concrete slab thickness increased by 6.4%, 7.6% and 4.0%, respectively, compared to that of the composite beam with 60 mm concrete slab thickness. In addition, the flexural stiffness of the GCC beams was increased by 6.2%, 12.7%, 19.2% and 26.1%, respectively, compared to that of the composite beam with 60 mm concrete slab thickness. The numerical results indicated that the bending bearing capacity and flexural rigidity of the composite beams were enhanced with the increase of concrete slab thickness.

## 5. Conclusions

To study the flexural performance of GCC beam containing interlayer, experimental research and finite element modelling of the composite beam were conducted. The finite element studies were performed to investigate the impacts of the glulam beam height, shear connector spacing, timber board thickness and concrete slab thickness on the structural properties of GCC beam.
(1)The experimental results demonstrated that the failure mechanism of the GCC beam was the combination of bend and tensile failure in the glulam beam. At the beginning of loading, the slip displacement was small because of the strong bonding effect at the concrete-timber interfacial region. When the loading was elevated to approximately 28% of the maximum load, with the applied loads increased, the attenuated composite actions at the interfacial region led to a rapid increase of the interface slips.(2)The finite element results showed that the numerical simulation could precisely predict the failure mode and the change characteristics of the composite beam in the loading process. The developed finite element model adequately predicted the load-deflection relationship and load-interface slip curve, particularly in the elasticity range. For the elastic-plastic stage, the predicted deflection of the GCC beam deviated from the test results due to the nonlinear bending performance. The elastic-plastic damage constitutive model of glulam needs to be optimized.(3)The parametric study results showed that with increase of the glulam beam height, the bending bearing capacity and flexural stiffness of the GCC beam were significantly improved. With the shear connectors spacing increased, the ultimate bearing capacity and bending stiffness of the composite beam were decreased. The existence of timber board had no significant effect on the flexural performance of the composite beams. The bending bearing capacity and flexural stiffness of the composite beams were enhanced with the increase of the concrete slab thickness.

## Figures and Tables

**Figure 1 materials-15-07998-f001:**
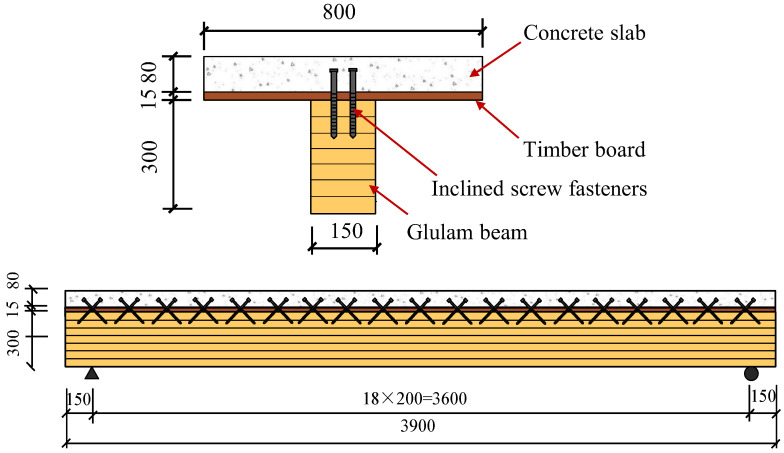
Dimensions of composite beam specimen GCCB.

**Figure 2 materials-15-07998-f002:**
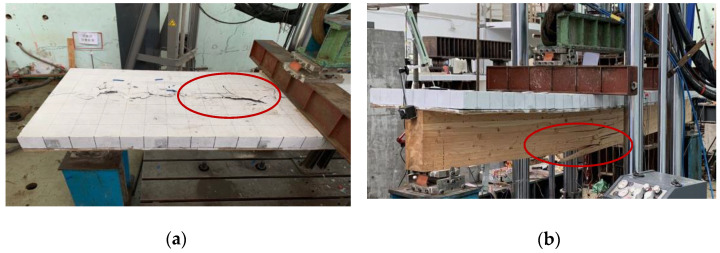
Failure mode of composite beam specimen. (**a**) Cone failure of concrete slab, (**b**) Tensile failure of glulam beam.

**Figure 3 materials-15-07998-f003:**
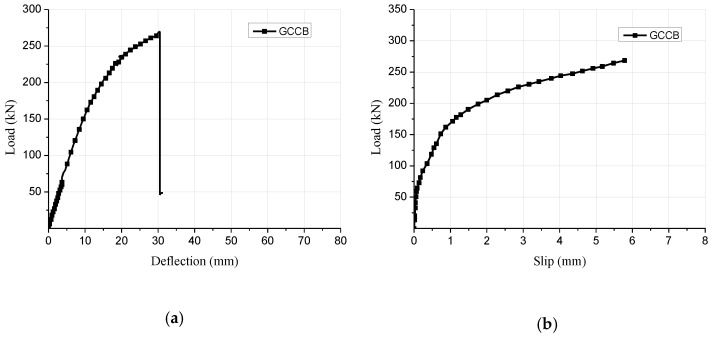
Test results of composite beam specimen. (**a**) Load-deflection curve, (**b**) Load-slip curve.

**Figure 4 materials-15-07998-f004:**
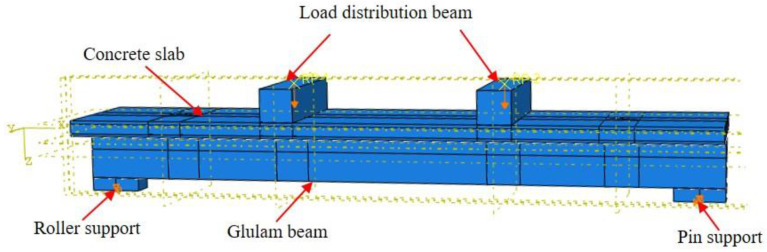
Model diagram of GCC beam after defining boundary and load conditions.

**Figure 5 materials-15-07998-f005:**
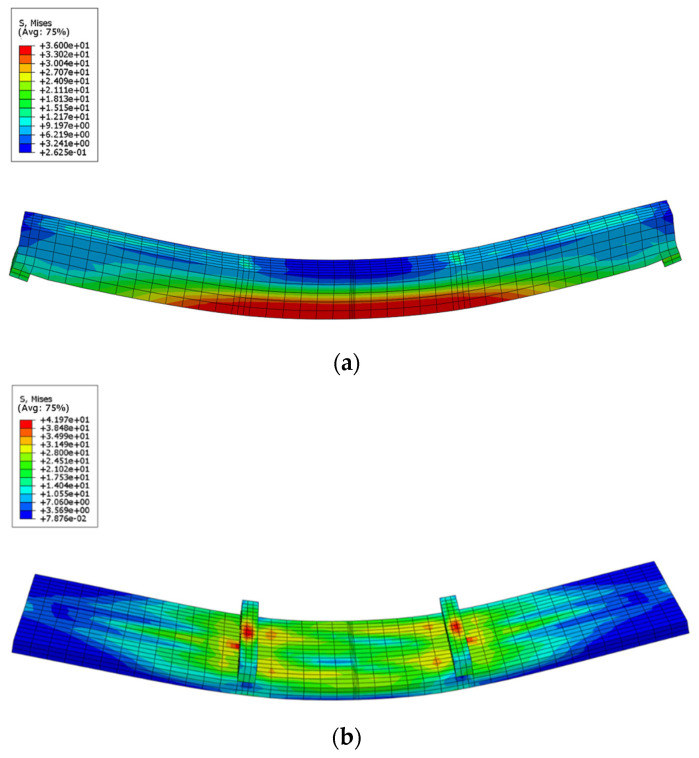
Stress distributions of GCC beam at maximum load. (**a**) Glulam beam, (**b**) Concrete slab.

**Figure 6 materials-15-07998-f006:**
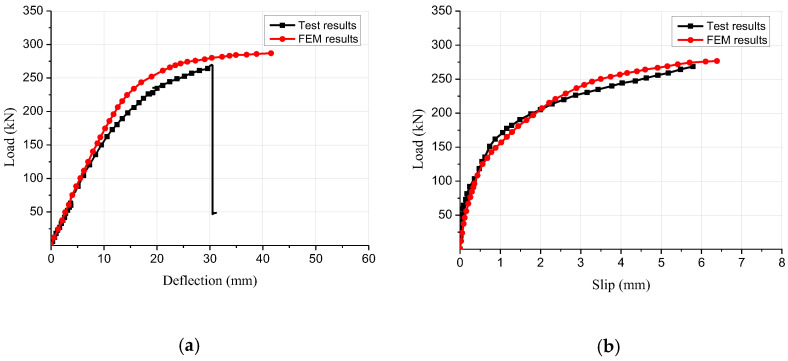
Comparison between the test and finite element results. (**a**) Load-deflection curve, (**b**) Load-slip curve.

**Figure 7 materials-15-07998-f007:**
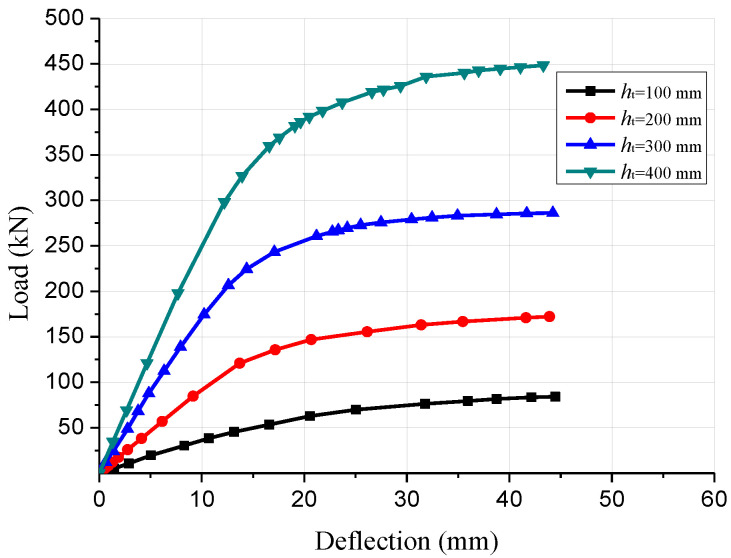
Comparison of load-deflection curves of composite beams with different beam heights.

**Figure 8 materials-15-07998-f008:**
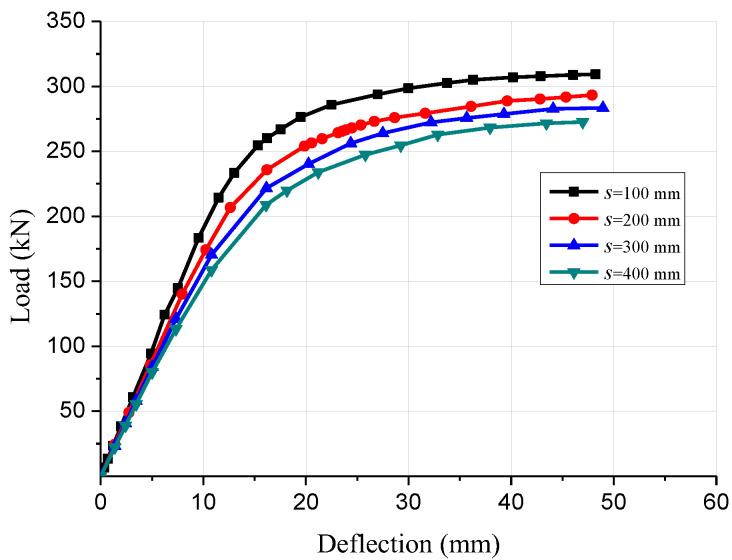
Comparison of load-deflection curves of composite beams with different screw connector spacing.

**Figure 9 materials-15-07998-f009:**
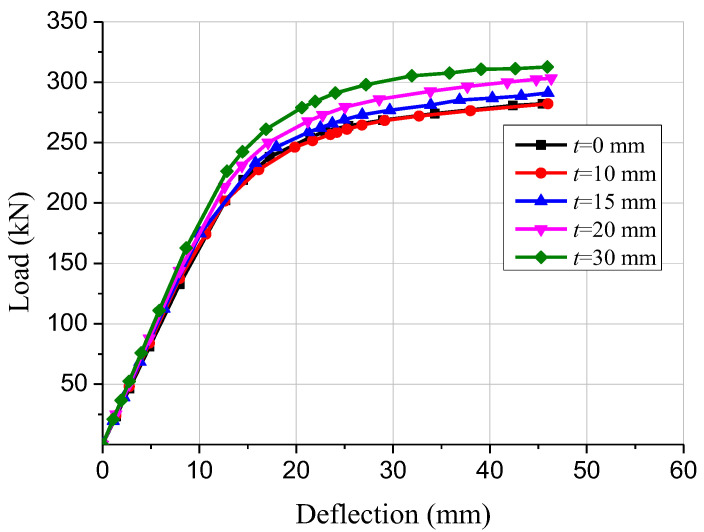
Comparison of load-deflection curves of composite beams with different timber board thicknesses.

**Figure 10 materials-15-07998-f010:**
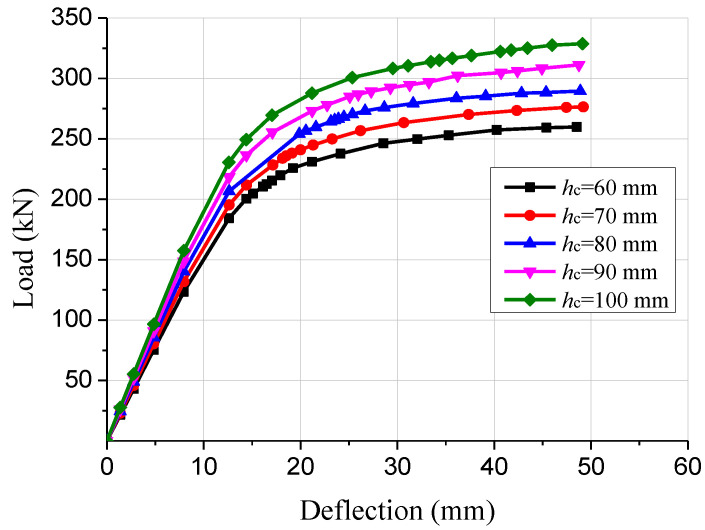
Comparison of load-deflection curves of composite beams with different concrete slab thicknesses.

## Data Availability

Not applicable.

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
