# Peer review of "Experimental and Finite Element Study on Bending Performance of Glulam-Concrete Composite Beam Reinforced with Timber Board"

_materials, 2022, doi:10.3390/ma15227998_

Round 1
Reviewer 1 Report
Is this application specific to your country or applied in many countries?
In abstract convey only about what is your work, what is done, what is outcome, this will be more perfect for readers.
Line 12: none-linear finite element model – Non linear is correct term.
Results are very obvious, no novelty, if you increase thickness obviously the resistance to load will increase, rather project what is the novelty in this work. Line 248 also confirms this, in this case the novelty of this work Is questionable.
Line 19: Then what is the use of this board, read first line and this line, both are contradictory.
glulam-concrete composite beam with timber board Add : reinforced with in title.
Is this word “The parametric analyses” clearly understood? Its used in many places in first few pages of article, kindly refrain from over using.
Last para of introduction need to be reframed and it should inform readers what exactly is done in this work clearly, in lucid way.
Line 87: only one specimen is constructed, is it representative sample to accept the results obtained is questionable.
Rewrite line 88,89 please, it is not providing proper meaning.
Materials and methods section is not clearly and well written.
Line 97: Why compressive strength is of importance to this work, I feel tensile and slip failure should have been studied in depth.
In title its given as numerical modelling but nowhere it exists in the article, kindly change it to FE modelling.
Why experimental research is coupled with FE work in this particular research is not clear anywhere in the article, it needs to be clearly defined.
In conclusion authors conveyed about slip failure, how and where its measured in this work?
Is the slip failure due to screw joint failure or any other mode?
Fig 3 shows deflection is too high, is there any codal provision available for this particular variety of beam?
Change section 3.1 heading properly.
three-dimensional FE models – What exactly is three dimensional FE model?
Line 139: give citation “Hill”
Fig 4(b) is it a slab? What is the perpendicular point element shown in that? Also, what stress is mentioned in that fig? von-misses?
FEA and FEM use clearly, use appropriately.
Is fig 7 logically correct? Please check.
Author Response
Manuscript ID: materials-1987645
Title: Experimental and numerical study on bending performance of glulam-concrete composite beam with timber board
Dear editor and reviewers,
Thank you for reviewing our manuscript and providing suggestions!
Our detailed point-by-point responses to the comments are as follows:
Response to the Reviewer 1:
Point 1. Is this application specific to your country or applied in many countries?
Response 1: GCC can be used as a refurbishment technique for old historical buildings in European cities, such as Leipzig in Germany. In the last 30 years, interest in GCC systems has increased, resulting in the construction of bridges (United States, New Zealand, Australia, Switzerland, Austria, and Scandinavian countries), upgrading of existing timber floors (Europe), and the construction of new buildings.
Point 2. In abstract convey only about what is your work, what is done, what is outcome, this will be more perfect for readers.
Response 2: We have revised the abstract to highlight what is your work, what is done, what is outcome.
Point 3. Line 12: none-linear finite element model – Non linear is correct term.
Response 3: We have revised “none-linear” as “non linear” in this sentence.
Point 4. Results are very obvious, no novelty, if you increase thickness obviously the resistance to load will increase, rather project what is the novelty in this work. Line 248 also confirms this, in this case the novelty of this work Is questionable.
Response 4: In different practical uses of glulam-concrete composite (GCC) beams, the timber boards are placed on top of the glulam members and used as the formwork for concrete slab casting. At present, the researches on the structural performances of glulam-concrete composite beams containing timber board interlayer are insufficient. The novelty of this paper was experimental research of the glulam-concrete composite beams containing timber board interlayer and developed the three-dimensional none linear finite element model. In addition, the parametric analyses were completed to explore the impacts of the glulam beam height, shear connector spacing, timber board interlayer thickness and concrete slab thickness on the flexural properties of composite beams.
Point 5. Line 19: Then what is the use of this board, read first line and this line, both are contradictory.
Response 5: We have revised this sentence. The timber boards were placed on top of the glulam members and used as the formwork for concrete slab casting. The flexural properties of composite beams were improved with the increase of the timber board thickness.
Point 6. glulam-concrete composite beam with timber board Add : reinforced with in title.
Response 6: We have revised the title and added “reinforced with” in the title.
Point 7. Is this word “The parametric analyses” clearly understood? Its used in many places in first few pages of article, kindly refrain from over using.
Response 7: We have revised the word “The parametric analyses” in first few pages of article.
Point 8. Last para of introduction need to be reframed and it should inform readers what exactly is done in this work clearly, in lucid way.
Response 8: We have revised the last para of introduction to exactly express the main work of this paper.
Point 9. Line 87: only one specimen is constructed, is it representative sample to accept the results obtained is questionable.
Response 9: In this paper, the experimental research of the glulam-concrete composite beams containing timber board interlayer was conducted. In addition, finite element simulation was conducted, and the experimental results and the finite element results are mutually verified. Similarly, a full-scale glulam-concrete composite beam was designed and constructed in references [1-3]
References
[1] Clouston P, Bathon L, Schreyer A. Shear and bending performance of a novel wood-concrete composite system. Journal of Structural Engineering, 2005, 131(9): 1404-1412.
[2] Hong W, Jiang Y, Hu X, et al. Experimental study and theoretical analysis of glulam-concrete composite beams connected with ductile shear connectors. Advances in Structural Engineering, 2019, 23(6): 1168-1178.
[3] Fragiacomo M. Experimental behaviour of a full-scale glulam-concrete composite floor with mechanical connectors. Materials and Structures. 2012, 45(11): 1717-1735.
Point 10. Rewrite line 88,89 please, it is not providing proper meaning.
Response 10: We have revised this sentence to provide proper meaning. In composite beam sample, the glulam beam was connected to the concrete slab using shear connectors (Fig. 1). The glulam beam was 150 mm wide, 300 mm high and 3900 mm long. The concrete slab was 80 mm thick and 800 mm wide.
Point 11. Materials and methods section is not clearly and well written.
Response 11: We have revised the materials and methods section to express clearly.
Point 12. Line 97: Why compressive strength is of importance to this work, I feel tensile and slip failure should have been studied in depth.
Response 12: This section mainly introduced the material properties of the timber and concrete. In addition, the constitutive model of the timber and concrete utilized in finite element model were determined according to the material test results. Therefore, the compressive strength of the concrete was used to determine the constitutive model of the concrete in finite element model.
Point 13. In title its given as numerical modelling but nowhere it exists in the article, kindly change it to FE modelling.
Response 13: We have revised the title to FE modelling.
Point 14. Why experimental research is coupled with FE work in this particular research is not clear anywhere in the article, it needs to be clearly defined.
Response 14: We have revised the last para of introduction to exactly express the main work of this paper.
Point 15. In conclusion authors conveyed about slip failure, how and where its measured in this work?
Response 15: We have added the detailed information about the measured method of the interface slip. For obtaining the relative slips between the concrete slab and the glulam beam, two displacement meters were mounted at the end of the GCC specimen.
Point 16. Is the slip failure due to screw joint failure or any other mode?
Response 16: As the bending load increasing, the relative slips at the interface increased due to the flexibility and bending deformation of the screw connectors under longitudinal shear forces.
Point 17. Fig 3 shows deflection is too high, is there any codal provision available for this particular variety of beam?
Response 17: The design of glulam-concrete beam is usually controlled by deflection. The corresponding mid-span deflection under serviceability limit state â–³SLS was based on GB 50005 [1], â–³SLS = l/250. We have added the explanation in this section.
Reference
[1] GB 50005, Standard for Design of Timber Structures, Standardization Administration of China, China, 2017.
Point 18. Change section 3.1 heading properly.
Response 18: We have revised section 3.1 heading to express properly.
Point 19. three-dimensional FE models – What exactly is three dimensional FE model?
Response 19: To sufficiently consider the effect of the failure mechanisms on the flexural behavior of GCC beam, the three-dimensional (3D) FE models were established. The 8-node solid element with reduced integration C3D8R was utilized to model the concrete slab, glulam beam and screw fasteners.
Point 20. Line 139: give citation “Hill”
Response 20: We have added citation “Hill” in this sentence.
Point 21. Fig 4(b) is it a slab? What is the perpendicular point element shown in that? Also, what stress is mentioned in that fig? von-misses?
Response 21: Fig 4(b) shows the von-misses distribution of the concrete slab in GCC beam at maximum load. The figure was showed the perpendicular to the middle face of the solid element of the concrete slab.
Point 22. FEA and FEM use clearly, use appropriately.
Response 22: We have revised this section to make a clear and appropriate expression.
Is fig 7 logically correct? Please check.
Response 23: We have checked the logical and correct of Fig. 7. With the screw fasteners spacing increased, the composite action between the glulam beam and the concrete slab was decreased. This resulted in the reduction of bending bearing capacity and flexural stiffness of the composite beams.

Reviewer 2 Report
The flexural performance of the glulam-concrete beam-slab composite system was investigated experimentally and numerically in this paper, which is well presented in the case study. However, the presentation of the paper needs further improvement by considering the following comments:
1. The judgment statement on the previous studies “At present, the researches ………are insufficient” need to be rewritten in a more technical way. Where the main difference/gap between the currently suggested study and the previously investigated studies in this field needs to be highlighted clearly.
2. The abstract could be more informative by providing the key results from the experimental and numerical works.
3. The introduction/conclusions need to highlight and discuss the limitation of this study.
4. In the introduction, please avoid the large bulk citations [21-31] (page 2, line 38), at least use small bulk citations (2-4 references) after highlighting the same common point(s) of these references.
5. In the introduction, the references’ numbering system needs to be rechecked and organized in sequence, see the reference’s number on page 2, lines 36 to 42 as an example.
6. Section 2.1, needs to explain and clarify technically why the inclined fasteners were used. Also, more details on the concrete mix and embedded reinforcement bars need to be highlighted. The loading test setup needs to be shown as well.
7. Section 3: For a better understanding of the modeling work, the boundary conditions of the FE model including the supports and loading scenario need to be shown in the figure and clearly highlighted.
8. Section 3: clearly clarify the definition of modeling the screw spacing, material, and/or type of interaction, since this particular matter was used as a parameter with different spacing in section 4.2
9. Fig. 4 must be improved, totally unreadable that the stresses values and contour lines distribution need to provide a bigger and clear picture for the FE models.
10. Page 10, line 184, correct figure number (Fig. 6)
Author Response
Manuscript ID: materials-1987645
Title: Experimental and numerical study on bending performance of glulam-concrete composite beam with timber board
Dear editor and reviewers,
Thank you for reviewing our manuscript and providing suggestions!
Our detailed point-by-point responses to the comments are as follows:
Response to the Reviewer 2:
Point 1. The judgment statement on the previous studies “At present, the researches ………are insufficient” need to be rewritten in a more technical way. Where the main difference/gap between the currently suggested study and the previously investigated studies in this field needs to be highlighted clearly.
Response 1: We have revised the judgment statement on the previous studies in this section to make a technical and appropriate expression.
Point 2. The abstract could be more informative by providing the key results from the experimental and numerical works.
Response 2: We have revised the abstract to provide the he key results from the experimental and numerical works.
Point 3. The introduction/conclusions need to highlight and discuss the limitation of this study.
Response 3: We have revised the introduction and conclusions to highlight and discuss the limitation of this study.
Point 4. In the introduction, please avoid the large bulk citations [21-31] (page 2, line 38), at least use small bulk citations (2-4 references) after highlighting the same common point(s) of these references.
Response 4: We have revised the bulk citations after highlighting the same common point(s) of these references in the introduction.
Point 5. In the introduction, the references’ numbering system needs to be rechecked and organized in sequence, see the reference’s number on page 2, lines 36 to 42 as an example.
Response 5: We have revised the references’ numbering system, and checked and organized in sequence.
Point 6. Section 2.1, needs to explain and clarify technically why the inclined fasteners were used. Also, more details on the concrete mix and embedded reinforcement bars need to be highlighted. The loading test setup needs to be shown as well.
Response 6: We have added the detailed information and explanations about why the inclined fasteners were used, embedded reinforcement bars in concrete slab and the loading test setup in the Section 2.1.
Point 7. Section 3: For a better understanding of the modeling work, the boundary conditions of the FE model including the supports and loading scenario need to be shown in the figure and clearly highlighted.
Response 7: We have added the figure to give a better explanation of the modeling work, the boundary conditions of the FE model including the supports and loading scenario in Section 3.
Point 8. Section 3: clearly clarify the definition of modeling the screw spacing, material, and/or type of interaction, since this particular matter was used as a parameter with different spacing in section 4.2.
Response 8: We have added the detailed information about the definition of modeling the screw spacing material properties, and type of interaction in Section 3.
Point 9. Fig. 4 must be improved, totally unreadable that the stresses values and contour lines distribution need to provide a bigger and clear picture for the FE models.
Response 9: We have revised the Fig. 4 to provide a bigger and clear picture for the FE models.
Point 10. Page 10, line 184, correct figure number (Fig. 6)
Response 10: We have revised and corrected the figure number in this sentence.

Round 2
Reviewer 2 Report
The given comments and suggestions are addressed in the revised paper and found acceptable.